# Strain-Induced Form Transition and Crystallization Behavior of the Transparent Polyamide

**DOI:** 10.3390/polym13071028

**Published:** 2021-03-26

**Authors:** Chenxu Zhou, Siyuan Dong, Ping Zhu, Jiguang Liu, Dujin Wang, Xia Dong

**Affiliations:** 1CAS Key Laboratory of Engineering Plastics, Beijing National Laboratory for Molecular Sciences, CAS Research/Education Center for Excellence in Molecular Sciences, Institute of Chemistry, Chinese Academy of Sciences, Beijing 100190, China; cxzhou@iccas.ac.cn (C.Z.); dpang2012@gmail.com (S.D.); pzhu507@iccas.ac.cn (P.Z.); djwang@iccas.ac.cn (D.W.); 2University of Chinese Academy of Sciences, Beijing 100049, China; 3College of Material Science and Engineering, Beijing Institute of Fashion Technology, Beijing 100029, China; j.liu@bift.edu.cn

**Keywords:** transparent polyamide, cold crystallization, crystallization form transition, strain-induced crystallization, extension

## Abstract

A transparent polyamide, poly(4,4′-aminocyclohexyl methylene dodecanedicarboxylamide) (PAPACM12), was studied and characterized by in situ wide-angle X-ray diffraction (WAXD) to establish the relationship between its crystallization behavior, crystalline form transition under external fields, and macroscopic properties. During the heating process, cold crystallization occurred and increased, and there was no form transition below the melting point. During the isothermal process, PAPACM12 exhibited the same crystalline structure as that during the heating process. The crystalline structure of PAPACM12 was attributed to *α*-form crystal, which is the stable form, according to the WAXD diffraction peaks of the conventional AABB-type polyamides. During stretching deformation, the crystal transition from *α*-form to *γ*-form and strain-induced crystallization were observed to contribute to the PAPACM12 with higher breaking strength and elongation. This study firstly determine the crystalline structure of transparent polyamides, and then the controlled strain-induced crystallization and transformation are demonstrated to be effective preparation methods for polyamides with high properties.

## 1. Introduction

Polyamides, with amide groups in the chain backbone to form inter- and intra-molecular hydrogen bonds, have high crystallinity, thermal, and mechanical properties [1,2,3,4,5]. Their unique performances make polyamides one of the five major engineering thermoplastics and enable them to be widely used in the textile, electronics, aerospace, automotive, and medical device industries, as well as other fields [6,7]. With the demand for some special applications, an increasing number of polyamide varieties are appearing [8,9]. The transparent polyamide is a novel kind of polyamide and is different from conventional, commercial ones, e.g., PA6 and PA66, in which the alicyclic or aromatic structures increase steric hindrance and reduce the density of hydrogen bonds and the crystallization ability [10,11]. Transparent polyamides are usually in the amorphous state or contain microcrystals whose size is smaller than the wavelength of visible light. Consequently, transparent polyamides show considerably high transparency, high mechanical strength, good rigidity, superior thermal stability, and good chemical resistance [12,13,14]. They are widely used in optical instruments, glasses, aerospace, and other fields. The microstructures determine the macroscopic properties. Therefore, it is necessary to study the microstructure of the transparent polyamide in order to improve its use in more fields.

Polyamides can be divided into AB and AABB types from the perspective of the repeating units or the monomers [15,16]. Among them, the AB-type polyamide is polymerized by either lactam or *ω*-amino acids, and the AABB-type polyamide is obtained by polymerization of dehydration condensation of dicarboxylic and diamine acid [17]. Transparent polyamides belong to AABB-type polyamides. Regarding the study of the microstructure of ABBB-type polyamides, as early as 1947, Bunn et al. [18] reported for the first time that the crystal phase of the *α*-form is the triclinic structure of PA66 and PA610 and found that in the X-ray diffraction (XRD) patterns, the *α*-form crystal has two characteristic diffraction peaks, denoted as (100) and (010)/(110) lattices, respectively. Among them, the *d*-space of (100) lattice in the diffraction pattern is 0.44 nm, which represents the space between adjacent chains within a hydrogen-bonded sheet. The overlap of the diffraction peaks of planes (010) and (110) with two close *d*-space values is about 0.37 nm. In 1959, Kinoshita [19] proposed the *γ*-form crystal of polyamides, which is generally formed under high temperatures and stretching fields. The structure of a *γ*-form crystal is a pseudo-hexagonal structure. In the X-ray diffraction pattern, there is only one characteristic diffraction peak.

Numerous studies also discussed the microstructure evolution of AABB-type polyamides under external fields. Brill [20] found that during the heating process of PA66, the *α*-form crystal and *γ*-form crystal could undergo a reversible transformation, which was defined as the Brill transition, and the corresponding temperature of transformation was defined as the Brill transition temperature (*T*_B_). Atkins et al. [2,21,22] studied the nature of crystal transition of AABB-type polyamides from the perspective of chain structures by transmission electron microscopy (TEM), XRD, and thermal analysis. They then proposed that the crystal transition only involves local atomic cooperative motion perpendicular to the chain axis and that the *T*_B_ is therefore independent of the length of the chain. Tashiro et al. [23] thoroughly studied the crystal transition by the infrared spectra of PA1010 under the temperature field and also observed a sequence of characteristic peaks related to crystal transition. By using variable temperature wide-angle X-ray diffraction (WAXD) and Fourier transformation infrared spectroscopy (FTIR), Yan et al. [24,25,26] studied crystal transition of a series of AABB-type polyamides, such as PA811, PA1011, PA1220, and PA1012, and determined the *T*_B_ of each polyamide. It was observed that a sequence of FTIR bands could be corresponding to different crystal forms. Moreover, many existing studies have shown crystal transition under the stretching field. Mo et al. [27] observed crystal transition in PA1212 under stretching by WAXD and confirmed that the stretching field plays a vital role in driving the transition from the *α*-form to *γ*-form crystal. Wang et al. [28] established the correlation between the microstructure and mechanical response of PA1012 by studying the microstructure evolution under uniaxial tensile deformation.

In recent years, the crystalline structure and crystal transition behavior of conventional polyamides have been widely studied [29,30,31], but the microstructure of transparent polyamides has not been sufficiently studied in the current literature. Referring to our previous research on PA1012, poly(4,4′-aminocyclohexyl methylene dodecanedicarboxylamide) (PAPACM12) with the same dicarboxylic acid structure (dodecanedioic acid) was selected for study in this paper. The crystallization behavior and form transition during the heating, isothermal, and stretching process were systematically studied by in situ wide-angle X-ray diffraction (WAXD) and modulated differential scanning calorimetry (MDSC). The study of microstructure and microstructure evolution under external fields of transparent polyamides contributes to the provision of a theoretical foundation for the development and application of such polyamides.

## 2. Materials and Methods

### 2.1. Materials

PAPACM12 was purchased from Evonik. The repeating unit of PAPACM12 according to its official technical datasheet is displayed in Figure 1. The thermal properties of PAPACM12 are shown in Table 1.

### 2.2. Preparation of Samples

The sample was dried in a vacuum oven at 100 °C for 12 h, melted, and then pressed under 60 bar and 255 °C for 3 min in a mold with a thickness of 0.3 mm to prepare for tensile tests by cutting it into dumbbell-shaped strips. Films with the dimensions of 25 × 3 × 0.3 mm were prepared for in situ X-ray diffraction measurements.

### 2.3. In Situ X-ray Diffraction Measurements

In situ X-ray diffraction measurements were performed at the Shanghai Synchrotron Radiation Facility (SSRF) with a radiation wavelength of 0.124 nm. A MAR CCD detector in the beamline BL14B1 with a resolution of 2048 × 2048 pixels was used to collect diffraction patterns. The distances of 187.2 and 1893.5 mm were used in the sample to detect space for wide-angle X-ray diffraction (WAXD) and small angle X-ray scatting (SAXS), respectively. All of the X-ray images were corrected for background scattering. Fit2D software was used to convert the two-dimensional (2D) X-ray patterns into one-dimensional (1D) curves.

Two procedures were used in X-ray diffractions measurements. The procedure under the stretching field: The stretching field was chosen to detect in situ microstructure evolution. Using the stretching speed of 20 μm/s in the Linkam TST350 (Houston, TX, USA) stretch stage, each exposure time was 20 s for both WAXD and SAXS measurements. The procedure under the temperature field: the film (thickness of about 1 mm) was wrapped with aluminum foil and isothermal for 5 min at 300 °C to eliminate thermal history; then, the film was cooled down to room temperature. The WAXD data were collected during the second heating process from room temperature to 300 °C in steps of 2 °C.

### 2.4. Tensile Testing

The samples with a dumbbell shape (25 × 3 × 0.3 mm) were tested by Instron 3365 (Norwood, MA, USA), under 23 °C and 30% relative humidity with a crosshead speed of 50 mm/min to obtain the tensile properties. All measurements were repeated at least 7 times.

### 2.5. Thermal Properties of Tensile Samples

The thermal properties were obtained by DSC and MDSC tests. Four types of samples were selected for testing. They were the pellets without being processed; the center position of the dumbbell-shaped strips without tensile strength; the center position of the samples that were stretched to the strain of 120%; and the center position of the samples that were stretched to the strain of 250% and were marked as original, undrawn, strain at 120%, and strain at 250%, respectively. The DSC program was isothermal for 5 min at 300 °C to eliminate the thermal history, then slowly cooled to 50 °C at a rate of 3 °C/min, and finally increased to 300 °C at a rate of 3 °C/min. The program of MDSC tests was the same as that of the DSC tests, except the rate was 10 °C/min, the modulation amplitude was ±1.79 °C, and the period was 60 s.

## 3. Results and Discussion

### 3.1. Cold Crystallization in Heating Process

The cold crystallization and melting processes of PAPACM12 were studied in our previous work [11]. The results of DSC show that cold crystallization occurs in the heating process and continues until the melting process begins. Moreover, the area of cold crystallization peak decreases as the heating rate decreases, and when the heating rate is less than 1 °C/min, the cold crystallization peak becomes less obvious. During the cooling process, the crystallization behavior of PAPACM12 is not obvious, and no crystallization peak appears.

The in situ WAXD could visually observe and record the changes in the aggregation structure of PAPACM12 during the heating process and determine the temperature range in which the change occurred. First, the cold crystallization temperature (*T*_cc_) was determined by DSC. According to the second heating curve of the DSC, the *T*_cc_ was 163.5 °C. The same cooling and heating procedure as that in the DSC test was carried out with the in situ WAXD. The 1D-integrated WAXD curves of PAPACM12 during the cooling process are shown in Appendix A. After eliminated the thermal history at 300 °C, there was only a broad diffraction peak during the cooling process, and the peak position shifted to a large angle with the temperature decreases. In accordance with the previous research on PA1012 [32,33,34], the 2*θ* of diffraction peak generated by the amorphous state was slightly less than that of the *γ*-form crystal. We observed that with the decrease in temperature, the diffraction peak at about 18° moved towards a higher degree, and a new diffraction peak at ca. 9° appeared, which may be related to the formation of the *γ*-form crystal. In order to accurately track the cold crystallization behavior of PAPACM12 in the heating process, the sample undergoing the same heat treatment as that in the DSC test was characterized by in situ WAXD. The 2D WAXD patterns (Appendix A) and the corresponding 1D-integrated curves (Figure 2) showed a diffuse ring at a low temperature without diffraction peaks, which resulted from the amorphous state in PAPACM12 in these states. When the temperature reached 160 °C, the diffuse ring turned into clear diffraction rings, which means that the ordered structure was formed by cold crystallization. However, there is no crystalline structure information about transparent polyamides reported to date. Based on the combination of our previous studies on the crystalline structure of the long-chain polyamide (LCPA) and literature on conventional polyamides [35,36,37,38], the crystal form of PAPACM12 formed by cold crystallization could be determined to be the *α*-form crystal of the AABB-type polyamide. Taking the 1D curve at 164 °C as the reference, the 2*θ*s of the diffraction peaks are at 13.9° and 15.3°, and the corresponding *d*-spaces are 0.51 and 0.46 nm, respectively, which is consistent with the conventional AABB-type polyamide. They are classified as the (100) and (010/110) lattices, respectively. For the conventional AABB-type polyamide, the *d*-space of the (100) lattice represents the distance between adjacent molecular chains within the hydrogen-bonded sheet and is 0.44 nm, and the *d*-space of the (010/110) lattice is 0.37 nm, which denotes the distance between adjacent hydrogen bond surfaces. The *d*-spaces of the corresponding diffraction peaks of PAPACM12 are 0.07 and 0.09 nm larger than those of the conventional AABB-type polyamide, respectively. This is because the repeating unit structure of PAPACM12 contains large alicyclic rings. Even if the rings adapt to the configuration of the boat or chair, the *d*-space should be slightly larger than that of the AABB-type polyamide with only aliphatic chains. In addition, the other two diffraction peaks (3.2° and 6.5°) also significantly changed with the increase in temperature, as shown in Figure 2. Table 2 shown the corresponding lattice assignment and *d*-space. The corresponding lattices are assigned to (001) and (002), respectively, and are both perpendicular to the *c*-axis. The corresponding *d*-space between the (001) and (002) lattices doubled in size. As the temperature approached *T*_m_, the diffraction peaks associated with the *α*-form crystal disappeared; that is, the crystalline structure formed by cold crystallization melted, and the macromolecular chains returned to the random state. However, during the heating process of PAPACM12, no crystal transition similar to conventional the AABB-type polyamide was observed, and only a similar *α*-form crystal was obtained.

### 3.2. Isothermal Crystallization

The crystallization rate of PAPACM12 is so slow that the crystallization behavior cannot be effectively observed during the conventional nonisothermal cooling process. When PAPACM12 experienced strong directional processing, such as injection molding and melt spinning, there was a very small exothermic peak in DSC testing near 190 °C; therefore, the *T*_c_ of PAPACM12 is 190 °C. The isothermal crystallization behavior of PAPACM12 was also studied with in situ WAXD. The 1D-integrated WAXD curves are shown in Figure 3. PAPACM12 was isothermal at 300 °C for 5 min to eliminate the thermal history and then cooled to 190 °C. Consequently, there was only one wide diffraction peak on the 1D curve, which means that there was mainly an amorphous structure in the PAPACM12. With the increase in isothermal time, new diffraction peaks appeared, which is similar to the cold crystallization behavior mentioned above. The corresponding lattice assignments and *d*-spaces of each diffraction peak are listed in Table 3. Similar to cold crystallization, a weak shoulder diffraction appeared near 11.5°, which has not yet been assigned to the corresponding lattices.

Above all, during the heating process and the isothermal process, the *α*-form crystal was dominant, but crystal transition was not observed.

### 3.3. Crystal Form Transition Induced by Strain

The microstructure evolution of the PAPACM12 system under heating/cooling treatment was also studied. However, it is well known that materials may be applied not only in the temperature field but also in the stretching field. In general, extension contributes to the orientation of macromolecular chains, but it is worth investigating whether the crystal transition occurs in PAPACM12 as in conventional AABB-type polyamides. Therefore, the microstructure evolution of PAPACM12 in the stretching field were characterized by in situ WAXD.

Figure 4 shown the engineering stress–strain curve of PAPACM12. According to the stress–strain relationship, the curve should be roughly divided into three parts, namely, the elastic stage (Region I); the necking (Region II), where the strain ranges from 25% to 175%; and the strain-hardening stage (Region III), with the strain in the range of from 200% to fracture strain. The untreated pellets and the samples under different strains (indicated by green dots in Figure 4) were selected for the MDSC test.

In MDSC tests, the reversible signal conveys information related to the change in heat capacity, such as crystal melting, glass transition, and solidification. The nonreversible thermal flow is induced by thermodynamics, such as crystallization, decomposition, volatilization, molecular relaxation, and chemical reaction. The MDSC test results for the four representative samples are shown in Figure 5 and Figure 6. Figure 5a,b show the curves of the reversible signal and the nonreversible signal during the first heating scans. It can be observed from Figure 5a that the *T*_g_ of the dumbbell strip gradually increased with the stretching strain but was always lower than that of the original pellet. The melting enthalpy showed the same result; that is, it increased with the increase in strain, which was due to strain-induced crystallization [38]. In addition, the melting peak changed from a single peak to double melting peaks. For the sample with a strain of 120%, the melting peak at the high temperature accounted for a greater proportion, while the melting peak at the low temperature dominated in the sample with a strain of 250%. This indicates that the crystalline structure was fractured into smaller crystalline grain in large-strain deformation. As shown in Figure 5b, the cold crystallization temperature decreased with the increase in strain. The cold crystallization temperature of the original pellet was about 175 °C, while the dumbbell strips generated by injection molding dropped to 145 °C. This may be due to the macromolecular chains with a certain degree of orientation behavior during injection molding. In addition, the degree of orientation increased with the increase in strain, which led to the further decrease in the cold crystallization temperature. Figure 6 shows the curves of the reversible signal and the nonreversible signal during cooling scans and the second heating scans. After annealed under 300 °C for 5 min to eliminate thermal history, the four samples all returned to the same state with the consistent thermodynamic properties.

2D WAXD patterns of PAPACM12 under stretching are shown in Figure 7. It can be seen that the original diffraction pattern contains a number of diffraction circulars. With the increase in strain, the diffraction pattern changed from circular to elliptical rings and finally evolved into two large arcs. This process is similar to the microstructure evolution of the conventional AABB polyamide during stretching. The Fit2D software was used to integrate 2D WAXD patterns of PAPACM12 in the meridional direction, and the range of the integrated angle was 45~135°. The corresponding 1D-integrated WAXD curves are exhibited in Figure 8, which shows that PAPACM12 in the initial state has two strong diffraction peaks, which proves that there is a certain ordered structure in the sample for injection molding. The *d*-spaces of the two diffraction peaks are 0.50 and 0.46 nm, respectively. According to the previous discussion, they are denoted as (100) and (010/110) lattices of the *α*-form crystal. With the increase in strain, the (100) lattice and (010/110) lattice gradually converge and eventually merge into a wider single diffraction peak, which could prove the formation of the *γ*-form crystal. The above crystal transition was observed in the 2D WAXD diffraction images in which the two diffraction rings gradually merge into a single diffraction ring with the increase in strain, and the crystal transition observed in the PAPACM12 system is consistent with the previous research on the AABB-type polyamide [27,28].

## 4. Conclusions

In this study, the crystalline structure and form transition of transparent polyamides during heating, isothermal, and stretching processes were sufficiently studied by in situ WAXD and MDSC. Under the heating process, cold crystallization was observed, and the diffraction peaks were similar to those of the *α*-form crystal of the conventional AABB-type polyamides. In addition, no crystal transition occurred during the entire heating process. During the isothermal process, PAPACM12 exhibited the same crystalline behavior as that during the heating process, which means that the formation of the *α*-form crystal forms favorably under different thermal conditions. Moreover, the crystalline transition from the *α*-form to the *γ*-form was observed under external deformation. This study determined the crystalline behavior and form transition of transparent polyamides and contributed to the provision of a theoretical foundation for the development and application of polyamides with high properties.

## Figures and Tables

**Figure 1 polymers-13-01028-f001:**
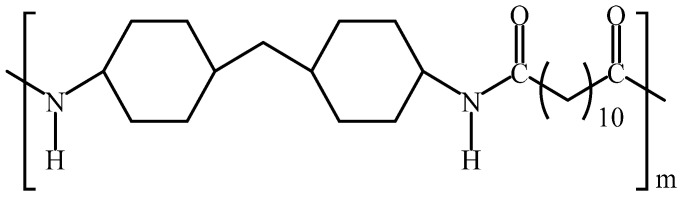
The repeating unit of poly(4,4′-aminocyclohexyl methylene dodecanedicarboxylamide) (PAPACM12).

**Figure 2 polymers-13-01028-f002:**
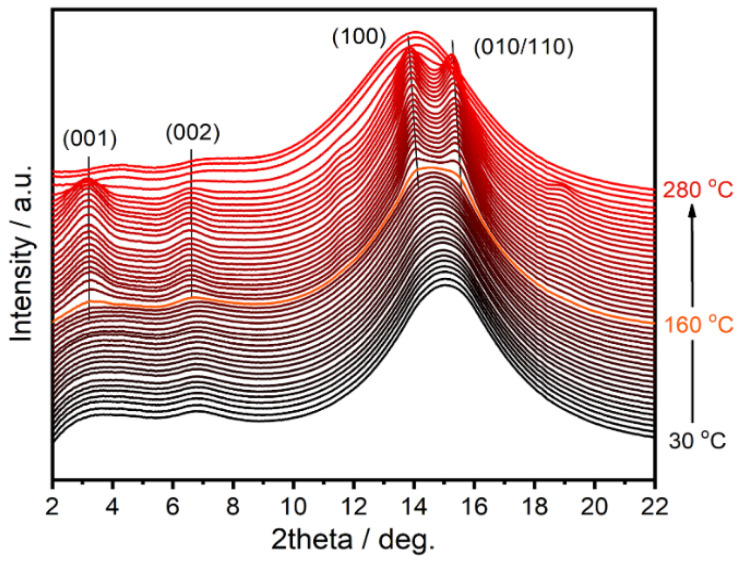
1D-integrated wide-angle X-ray diffraction (WAXD) curves of PAPACM12 in the heating process.

**Figure 3 polymers-13-01028-f003:**
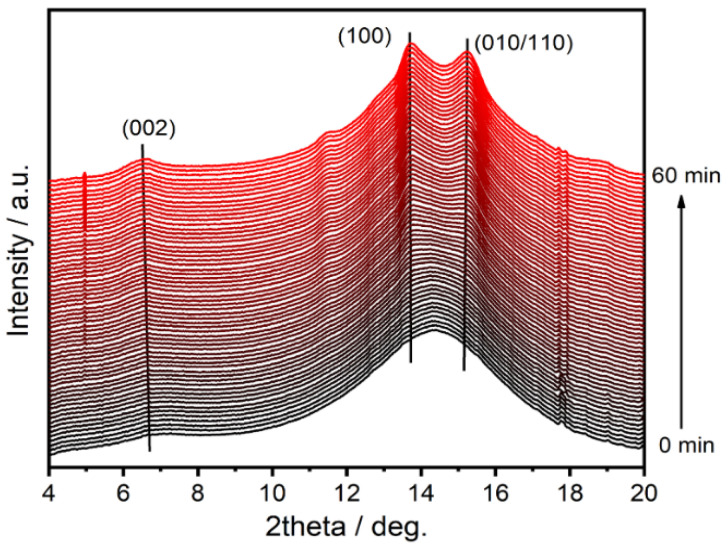
1D-integrated WAXD profiles of PAPACM12 during isothermal crystallization process under 190 °C.

**Figure 4 polymers-13-01028-f004:**
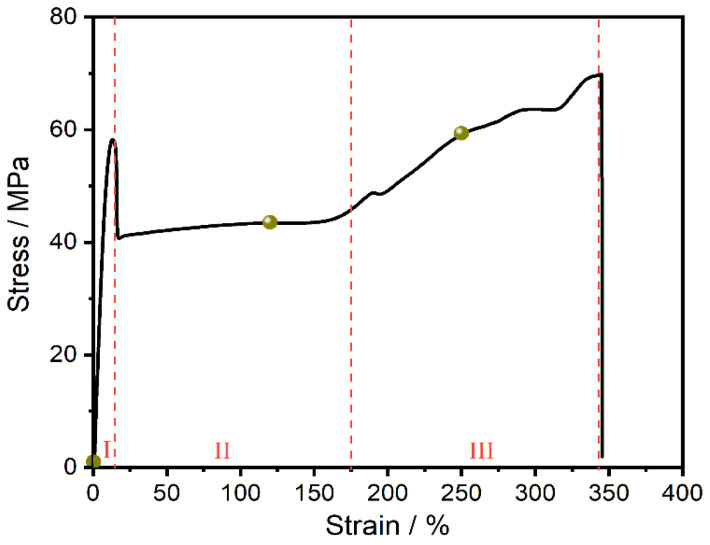
Engineering stress–strain curve of PAPACM12.

**Figure 5 polymers-13-01028-f005:**
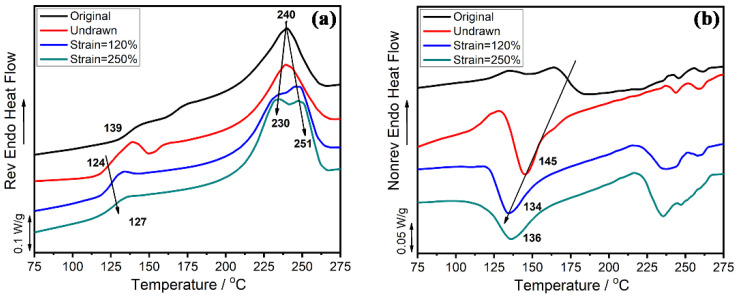
The first heating curves of PAPACM12 at different strains with modulated differential scanning calorimetry (MDSC): (**a**) reversible signal, (**b**) nonreversible signal.

**Figure 6 polymers-13-01028-f006:**
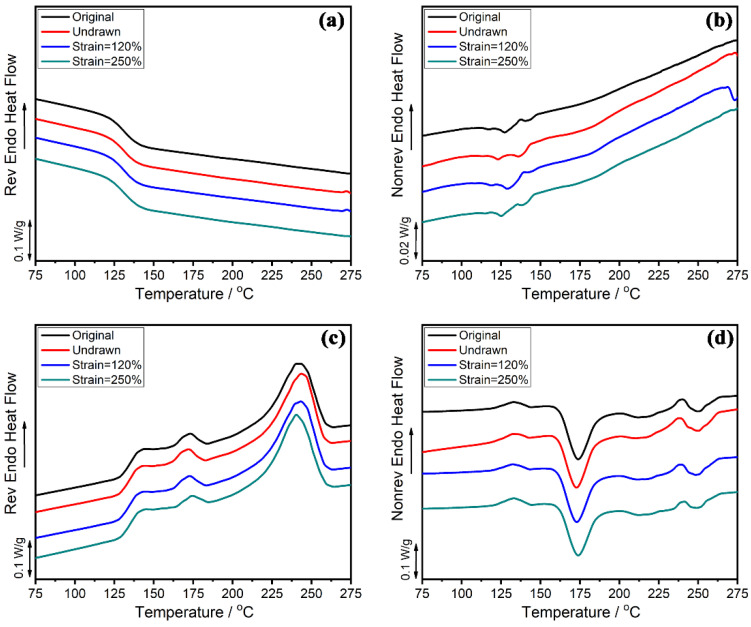
MDSC curves of PAPACM12 under different strain during cooling: (**a**) reversible signal, (**b**) nonreversible signal; second heating curves: (**c**) reversible signal, (**d**) nonreversible signal.

**Figure 7 polymers-13-01028-f007:**
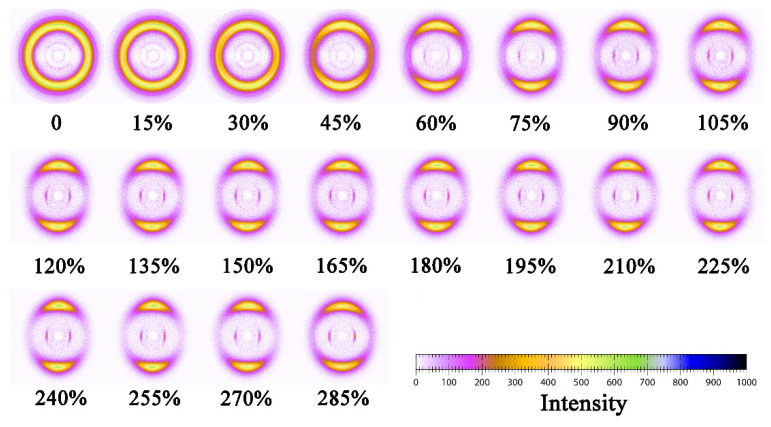
2D WAXD patterns of PAPACM12 under stretching with strain marked. Notes: stretching was performed along the horizontal direction.

**Figure 8 polymers-13-01028-f008:**
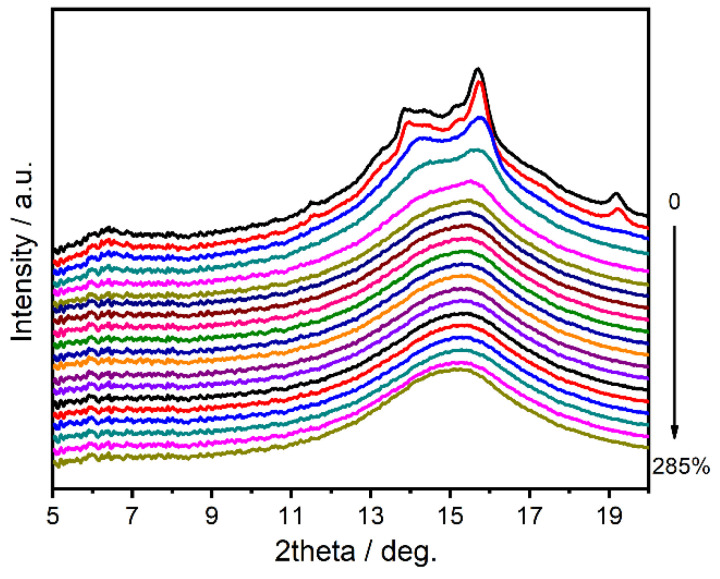
1D-integrated WAXD curves in the meridional direction of PAPACM12.

**Table 1 polymers-13-01028-t001:** The thermal properties of PAPACM12.

Polyamide	The Fastest Mass Loss Temperature ^a^ *T*_peak_/°C	Glass Transition Temperature ^b^ *T*_g_/°C	Cold Crystallization Temperature ^b^ *T*_cc_/°C	Melting Temperature ^b^ *T*_m_/°C
PAPACM12	482.1	136.4	174.2	244.7

^a^*T*_peak_ is determined by thermal gravimetric analysis (TGA, PE Pyris 1) (Appendix A). ^b^
*T*_g_, *T*_cc_, and *T*_m_ are determined by differential scanning calorimetry (DSC, TA Q2000) (Appendix A).

**Table 2 polymers-13-01028-t002:** The position of diffraction peak and corresponding *d*-space in 1D WAXD curves under 164 °C.

2*θ* (°)	*d*-Space/nm	Assigned Lattice Plane
3.2	2.18	(001)
6.5	1.09	(002)
13.9	0.51	(100)
15.3	0.46	(010)/(110)

**Table 3 polymers-13-01028-t003:** The position of the diffraction peak and corresponding *d*-space in 1D WAXD curves at 190 °C for 60 min.

2*θ* (°)	*d*-Space/nm	Assigned Lattice
6.5	1.09	(002)
13.7	0.52	(100)
15.2	0.47	(010)/(110)

## Data Availability

The data presented in this study are available on request from the corresponding author.

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
