# Peer review of "Strain-Induced Form Transition and Crystallization Behavior of the Transparent Polyamide"

_polymers, 2021, doi:10.3390/polym13071028_

Round 1

Reviewer 1 Report

Paper is well organized and contains sound scientific results. Before publisjing, some points should be corrected, namely:

1) Table 1. Temperatures designation should be decoded and explained.

2) Section 2. Tensile experiment should be described in detail here.

3) Please also describe why you select namely tensile test (not comression, bending and so on) for strain inducing.

Reviewer 2 Report

This manuscript deals with the crystallization behavior and solid-solid  phase transition of poly(4,4’-aminocyclohexyl methylene  dodecanedicarboxylamide) (PAPACM12). Based on X ray diffraction, it has been shown that by cold and isothermal crystallization only the α phase is formed; this polymorph transforms in the É£ phase by applying a controlled strain. The article is suitable for publication as it is, with the only advice to mention the disadvantage of X-Ray interpretation for very small crystals.
